# DanceQ: High-performance library for number conserving bases

Robin Schäfer[1, *] and David J. Luitz[2, †]

[1]*Department of Physics, Boston University, Boston, Massachusetts 02215, USA*
[2]*Institute of Physics, University of Bonn, Nussallee 12, 53115 Bonn, Germany*

(Dated: July 23, 2024)

The complexity of quantum many-body problems scales exponentially with the size of the system, rendering any finite size scaling analysis a formidable challenge. This is particularly true for methods based on the full representation of the wave function, where one simply accepts the enormous Hilbert space dimensions and performs linear algebra operations, e.g., for finding the ground state of the Hamiltonian. If the system satisfies an underlying symmetry where an operator with degenerate spectrum commutes with the Hamiltonian, it can be block-diagonalized, thus reducing the complexity at the expense of additional bookkeeping. At the most basic level, required for Krylov space techniques (like the Lanczos algorithm) it is necessary to implement a matrix-vector product of a block of the Hamiltonian with arbitrary block-wavefunctions, potentially without holding the Hamiltonian block in memory. An efficient implementation of this operation requires the calculation of the position of an arbitrary basis vector in the canonical ordering of the basis of the block. We present here an elegant and powerful, multi-dimensional approach to this problem for the $U(1)$ symmetry appearing in problems with particle number conservation. Our divide-and-conquer algorithm uses multiple subsystems and hence generalizes previous approaches to make them scalable. In addition to the theoretical presentation of our algorithm, we provide DanceQ, a flexible and modern – header only – `C++20` implementation to manipulate, enumerate, and map to its index any basis state in a given particle number sector as open source software under https://DanceQ.gitlab.io/danceq.

## I. INTRODUCTION

For quantum many-body problems, the size of the Hilbert space grows exponentially with the size of the system. Since there are only a handful of exactly solvable and non-trivial interacting models [1–3], we have to rely on approximations of various degrees of sophistication [4–6] and numerical methods [7–10] to study interacting systems. Numerical approaches started to pick up momentum in the 1950s with the increasing availability of computational power motivating new algorithmic developments of particular relevance for condensed matter physics [11–14] and the subsequent birth of computational physics [15–19]. In particular, it became possible to compute the spectrum of small, but generic interacting many-body systems [20]. In 1958, for example, R. Orbach used an *IBM 701* to compute eigenvalues of a chain of ten spins [21]. The success of numerical simulations of one-dimensional systems [22–24] quickly swapped over to higher dimensions [25–27] due to the exponentially growth in computer power [28, 29]. This steady growth of computer power makes it possible today to compute ground states for magnetic systems containing up to 50 spin-1/2 particles [30–32] with total Hilbert space dimensions exceeding $10^{15}$.

Brute force methods directly tackling the exponentially increasing complexity of the Hilbert space by encoding all details of the wave function fall under the category of *exact diagonalization*. Compared to other computational techniques frequently used in the field [8, 9], their advantages are their wide applicability and unbiased nature, in particular for cases where wave functions are strongly entangled. Naturally, the exponential growth in complexity of the problem imposed by quantum mechanics is a major obstacle for the solution of larger systems, which in turn are required for a valid finite size scaling analysis to address the thermodynamic limit. Therefore, any reduction of the problem – such as by exploiting symmetries to block-diagonalize the Hamiltonian – should be employed. Besides lattice symmetries that depend on the precise geometry of the problem [32–35], more intrinsic properties independent of the spatial structure play a crucial role in many physical systems. One such property is the conservation of the particle number, leading to the simplest scheme for block-diagonalization which is the focus of this work. Number conservation naturally arises in simple tight-binding type models [36] and in magnetic spin systems where the equivalent symmetry is related to the total magnetization. While organizing and managing the basis states may seem straightforward at first glance, the task becomes increasingly complex as the number of particles and system size grow, as outlined below [7, 37, 38].

In this work, we present an efficient algorithm to handle and organize the basis states of number-conserving systems based on a general divide-and-conquer approach. Specifically, consider a system consisting of $L$ individual sites, where each site hosts a quantum degree of freedom (qudit) with a local Hilbert space dimension $Q$ with the basis states $|\sigma_i\rangle \in \{|0\rangle, |1\rangle, \ldots, |Q-1\rangle\}$. The total *particle number* of the many-body system is

$$n = \sum_{i=1}^{L} \sigma_i \,. \tag{1}$$

* rschaefe@bu.edu
† david.luitz@uni-bonn.de

In the language of bosons, $\sigma_i$ refers to the number of particles located at a discrete lattice site where the maximal number of particles per site is $Q-1$. Alternatively, we can think of the total magnetization of $L$ spin-$S$ instances with $2S+1=Q$. In this scenario, the total particle number is replaced by the total magnetization along the $z$-axis:

$$S_{\text{tot}}^z = \sum_{i=1}^{L} S_i^z \,. \tag{2}$$

Our algorithm efficiently manages the comprehensive organization and manipulation of these basis states, which is a crucial element for methods based on exact diagonalization.

Naively organizing all basis states with a fixed particle number in a list, hash tables [39], or in lexicographical order [40] quickly suffers from an exponentially increasing overhead. For example, considering $L = 32$ spin-1/2 particles at half filling allocates approximately $18\,\text{GiB}$ of additional memory, which may be needed elsewhere. To overcome this barrier and making larger systems accessible, Lin [7, 37] proposed the decomposition into two subsystems reducing the memory consumption of lookup tables from $\mathcal{O}\left(e^L\right)$ to $\mathcal{O}\left(e^{L/2}\right)$ (a high performance implementation is for example provided in Ref. [41]). However, with the advancement of technology and massive parallelization over the past decades, even larger systems have become accessible, necessitating an even greater compression of lookup tables in massively parallel codes. Inspired by Lin's approach, we have generalized this idea into a "divide-and-conquer" ansatz allowing the decomposition into $N$ subsystems yielding a reduction of $\mathcal{O}\left(e^{L/N}\right)$.

The newly achieved reduction is extremely important for large, dilute systems and for massively parallel *sparse* and *matrix-free* applications. In the former case, the critical bottleneck in state enumeration can be circumvented, while in the latter case the reduced required storage for index lookup makes it possible that each worker, dealing with a part of the Hilbert space, can hold thread-local lookup tables for fast and synchronization-free state-to-index mapping (details below).

We have integrated our multi-dimensional search algorithm into a modern `C++20` implementation — **DanceQ** [42, 43] available as open source software under https://gitlab.com/DanceQ/danceq — capable of generating arbitrary particle number preserving Hamiltonians for arbitrary $Q$. Our implementation features an `MPI`-based, matrix-free version of the Lanczos algorithm for ground-state searches [11], along with a frontend for advanced parallel libraries such as `Petsc` [44, 45] and `Slepc` [46, 47]. It provides a user-friendly interface ready to exploit the full potential of current high performance computing facilitates.

While our motivation is driven by the application to physical systems, the problem to efficiently compute a lexicographic one-to-one mapping is a well-known problem in computer science and combinatorics and referred

to as *enumerative encoding* [48, 49]. Our generic algorithm, and previous variants [7, 37, 50], can be derived from the general ansatz provided by Cover in 1973 [49], a link we establish in Sec. III D.

This paper is organized as follows: In Sec. II, we introduce the problem and the desired features needed to efficiently construct an operator acting on a particle number sector. Then, Sec. III focusses on our divide-and-conquer algorithm. We start by discussing a concrete example followed by the general algorithm. For further clarification, we present two important limits: Lin's original proposal [7, 37] with two subsystems and the limit dividing the system into $L$ subsystems containing a single site each [50]. Next, we refer to Cover's formulation [49] and end by discussing the lookup tables that are used within each subsystem. In Sec. III F, we explain how to efficiently store compressed versions of local operators, such as the Hamiltonian using sparse tensor storage. Sec. IV analyzes the performance and determines the optimal choice of the partitioning. Lastly, Sec. V summarizes our work.

A detailed description of how to use the code can be found in the documentation [43] along with the source code.

## II. OVERVIEW

This section briefly introduces the problem and the most important features of the code necessary to carry out a (matrix-free) matrix-vector product using parallel working threads.

### A. The problem

We begin with a single lattice site with $Q$ degrees of freedom, corresponding to a local Hilbert space dimension $Q$ and label the basis states by $|0\rangle, \ldots, |Q-1\rangle$. A product state of the full system composed of $L$ such sites is represented by the tensor product of basis states of the individual sites:

$$|\vec{\sigma}\rangle := \bigotimes_{i=1}^{L} |\sigma_i\rangle = |\sigma_1; \ldots; \sigma_L\rangle \tag{3}$$

$$\text{with } |\sigma_i\rangle \in \{|0\rangle, \ldots, |Q-1\rangle\} \tag{4}$$

This induces a total Hilbert space dimension of $Q^L$ for the full system of $L$ sites. For systems with particle number conservation, it is useful to systematically focus on states with a fixed particle number $n \in \{0, \ldots, (Q-1)L\}$:

$$\hat{n}|\sigma_1; \ldots; \sigma_L\rangle = \left(\sum_{i=1}^{L} \sigma_i\right) |\vec{\sigma}\rangle = n|\vec{\sigma}\rangle \tag{5}$$

The number of such basis states with fixed particle number $n$ is the dimension of the corresponding symmetry sector. For the case $Q = 2$ it is well known that the

number of basis states for $n$ particles on a total of $L$ is given by

$$D_{Q=2}(L,n) = \binom{L}{n}, \qquad (6)$$

since it corresponds to the number of distinct ways to distribute $n$ indistinguishable items (particles) on $L$ sites.

For the general case with arbitrary $Q \geq 2$, the dimension of the symmetry sector with $n$ particles on $L$ sites is given by

$$D_Q(L,n) = \sum_{k=0}^{\lfloor n/q \rfloor} (-1)^k \binom{L}{k} \binom{L-1+n-qk}{L-1}, \quad (7)$$

where $\lfloor \bullet \rfloor = \text{floor}(\bullet)$ is the lower Gauss bracket defined by the integer part of the argument. We provide an explicit elementary derivation of this result in Sec. A in the appendix. The result in Eq. (7) was brought forward in the context of exact diagonalization for spin systems in Ref. [51] (cf. Eqs. (11), (12) in [51]), where it was also traced back to early work by De Moivre. It has also been used to enumerate permanents in bosonic systems [52, 53] and was derived in an alternative way by Ref. [54] in appendix B.

In order to represent wave functions as vectors and operators as matrices on a computer, it is necessary to impose a *canonical order* of all $D_Q(L,n)$ basis states in a symmetry sector. This order can be arbitrary but must not be changed during the calculation. In condensed matter physics and chemistry, the regime of interest is typically large $L$ and $n$ and hence the goal is to obtain for example the low energy behavior of a model Hamiltonian, i.e. to calculate the ground state in a given particle number sector. This can be achieved using Krylov space techniques like the Lanczos algorithm [11, 12, 55], for which it is sufficient to be able to calculate the action of the Hamiltonian $H$ on an arbitrary many-body wave-function $H|\psi\rangle$, without storing the (large and usually very sparse) matrix representation of $H$.

To carry out the matrix vector product $H|\psi\rangle$ efficiently, it is crucial to be able to access basis states by their index, i.e. the forward map

$$\text{index} \rightarrow |\sigma_0, \sigma_1, \ldots, \sigma_{L-1}\rangle, \qquad (8)$$

as well as to retrieve the index of a given basis state, i.e. the reverse map

$$|\sigma_0, \sigma_1, \ldots, \sigma_{L-1}\rangle \rightarrow \text{index}, \qquad (9)$$

because the action of an off-diagonal matrix element of $H$ effectively changes the basis state and we have to determine the corresponding row index in the result vector. This task can in principle be fulfilled by a lookup table of size $D_Q(L,n)$ for the forward lookup (state from index) and a lookup table of size $Q^L$ for the reverse lookup (index to state), but this requires an exponential memory overhead (by far exceeding the memory needed for storing wave-functions) and the goal of divide-and-conquer

approaches as the one presented here is precisely to avoid this overhead. Note that even though a forward lookup table of size $D_Q(L,n)$ for the map

$$\text{index} \rightarrow |\sigma_0, \sigma_1, \ldots \sigma_{L-1}\rangle \qquad (10)$$

can in principle be stored (its size is the size of a wave-function in the $n$ particle sector), a simple binary search in this table for reverse lookup of cost $O(\ln D_Q(L,n))$ (memory access) is prohibitively expensive.

## B. The code

The DanceQ library efficiently generates all basis states in any given particle number sector to represent the corresponding block of an operator in this sector.

A parallelized program requires three different functions:

(i) `get_index(|σ⃗⟩)`
Maps a valid (correct particle number) basis state $|\vec{\sigma}\rangle$ to a unique index in the canonical basis order ranging from 0 to $D_Q(L,n) - 1$.

(ii) `increment(|σ⃗⟩)`
Returns the next valid basis state in the canonical basis order such that
`get_index(|σ⃗′⟩) = get_index(|σ⃗⟩)+1`
with $|\vec{\sigma}\prime\rangle = $ `increment(|σ⃗⟩)`.

(iii) `get_state(k)`
The reversed mapping of function (i), i.e. it returns the basis state with a given index $k$ in the canonical basis order, such that
`get_index(get_state(k)) = k`.

The matrix free matrix vector product $H|\psi\rangle$ for any wave function $|\psi\rangle$ with coefficients $\langle \vec{\sigma} | \psi \rangle$ is generated by iterating over all basis states of the particle number sector using function (ii). Function (iii) is only executed to obtain the initial state, which becomes non-trivial in parallel programs, in which each worker transverses a different segment of the basis. An operator in matrix form is obtained by applying it to a specific state defining the current row. Then, function (i) is applied to obtain the respective column indices. Pictorially, a parallel program would split the Hamiltonian matrix into rectangular blocks (left) and the input wave function vector (center) and output vector (right) into subvectors like this:

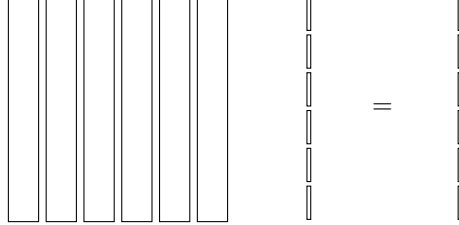

While functions (i) and (ii) are frequently executed during the construction of the operator, the function (iii) is only used once per worker process. Each worker handling one consecutive part of the basis (consecutive rows in the input wave function) executes function (iii) in the beginning to access its part. Pseudo codes for each function are attached in the Sec. B in the appendix.

## III. THE ALGORITHM

The key idea to efficiently handle fixed-$n$ basis states $|\sigma_1, \sigma_2 \ldots \sigma_L\rangle$ and to overcome the exponential memory overhead is a "divide-and-conquer" ansatz where we divide the whole system of size $L$ into a partition with $N$ subsystems.

$$\underbrace{\circ\circ\circ\circ}_{P_0}\underbrace{\circ\circ\circ\circ}_{P_1}\underbrace{\circ\circ\circ\circ}_{P_2}\underbrace{\circ\circ\circ\circ}_{P_3}\cdots\underbrace{\circ\circ\circ\circ}_{P_{N-1}}$$

We note that despite the pictorial representation in one dimension, this technique can be used for any geometry of the physical system. It is however crucial to introduce an order of the sites in the system and this is reflected in the partitioning. The index of a specific state is obtained by adding up contributions from the individual subsystems as we will elaborate in the following.

We label the subsystems by $P_k$, where $k$ indicates the $k$-*th* part. Because our basis states $|\vec{\sigma}\rangle$ are simple product states of single site states, they are also products of the individual subsystem states:

$$|\vec{\sigma}\rangle = |\vec{\sigma^{(0)}}\rangle_{P_0} \otimes |\vec{\sigma}^{(1)}\rangle_{P_1} \otimes \cdots \otimes |\vec{\sigma}^{(N-1)}\rangle_{P_{N-1}} \quad (11)$$

The remainder of this section is structured as follows. We begin by discussing an illustrative example using three subsystems which is depicted in Fig. 1, Fig. 2, and Fig. 3. After the example, we discuss the generalization of the algorithm to $N$ subsystems. To connect to prior work, we present two important limits including Lin's original approach [7] which corresponds to the case of two subsystems and the limit of $N = L$ subsystems [50] of size one which both follow trivially from the general formalism.

### A. A concrete example

To understand the general idea of the multidimensional index lookup, it is useful to begin with an illustrative example. We consider a system of $L = 9$ sites with $Q = 2$ and a total of $n = 4$ particles, split into $N = 3$ subsystems which we label $A \equiv P_0$, $B \equiv P_1$, $C \equiv P_2$ for simplicity. The total number of states is $D_2(9, 4) = 126$. We take all systems to have the same length $L_A = L_B = L_C = 3$. While the allowed states of the total system are limited by the fixed particle number $n = 4$, each subsystem can

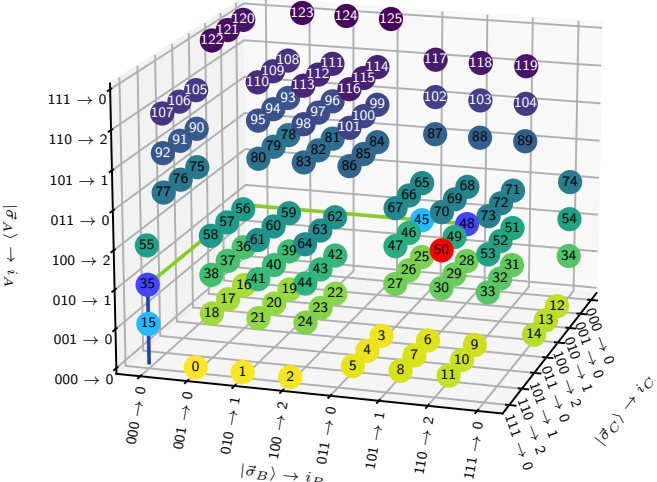

FIG. 1. Illustration of the three dimensional search structure emerging from three subsystems, $A$, $B$, and $C$. Each subsystem consists of three sites, the full system has hence length $L = 9$ and the figure shows all $D_2(9, 4) = 126$ basis states for $Q = 2$ and $n = 4$ particles. (Light) blue and red colored balls connected by blue and green lines indicate the path to finding the index 50 of the state $|\vec{\sigma}\rangle = |010\rangle_A \otimes |101\rangle_B \otimes |100\rangle_C$ using the divide and conquer approach. We start with the state $|010\rangle_A$ on the $A$ subsystem. It is in the $n_A = 1$ block, which has an offset of 15 (light blue). Within this block, $|010\rangle_A$ has the index $i_A = 1$ and the stride of this block is stride$_A = 20$. Hence, the contribution $c_A$ to the final index is $c_A = $ offset$_A + i_A$stride$_A = 35$ (dark blue). The state on subsystem $B$, $|101\rangle_B$ has the index $i_B = 1$ in the $n_B = 2$ block with offset$_B = 10$ and stride$_B = 3$, yielding $c_B = 13$. This brings us to the index $c_A + c_B = 48$ (dark blue) and by finally considering $|100\rangle_C$ with index 2 in the $n_C = 1$ block (with offset 0 and stride 1 since this is the last subsystem), we get $c_C = 1$. Hence, the final result for the desired index is $c_A + c_B + c_C = 50$ (red ball).

in principle be in any of the $Q^{L_A} = 2^3 = 8$ states: $|000\rangle$, $|001\rangle$, $|010\rangle$, $|100\rangle$, $|011\rangle$, $|101\rangle$, $|110\rangle$, and $|111\rangle$, however, if $A$ is in state $|111\rangle$, $B$ can only be in $|000\rangle$, $|001\rangle$, $|010\rangle$, or $|100\rangle$ due to the global constraint and it is precisely this kind of restriction which we need to deal with when enumerating all valid states.

If we order the states in each subsystem by the number of particles in the subsystem (the above list is already ordered in this way), and plot the subsystem states in the $x$, $y$, and $z$ axes of the 3d plot in Fig. 1, we can enumerate all allowed states $|\vec{\sigma}\rangle_A \otimes |\vec{\sigma}\rangle_B \otimes |\vec{\sigma}\rangle_C$ and draw a point at the appropriate position of the coordinate system along with the corresponding index of the obtained state in the full basis. The emerging structure in Fig. 1 are dense blocks of states, while the voids between the blocks correspond to states which do not fulfill the global constraint $n = 4$. Each dense cuboid block is made from all states with fixed subsystem particle numbers, i.e. with fixed $(n_A, n_B, n_C)$. This structure highlights the importance of ordering the subsystem bases by particle numbers and makes the key concept clear: We now have a structure

of dense blocks of states, in which we can efficiently retrieve the index of any state if we are able to skip all prior blocks in a straightforward way. To do this, we first explain how we organize the global basis states, i.e. in which sequence we walk through the structure shown in Fig. 1.

We make the choice to first increment the state of the $C$ subsystem, keeping the order of states organized by the subsystem particle number $n_C$ as pointed out before. Once the subsystem state $|\vec{\sigma}_C\rangle$ has cycled through all possible states (which sometimes are single choices as visible for states 0, 1, and 2 in Fig. 1) for fixed states on $A$ and $B$, we increment the $B$ state and only once also $B$ has exhausted its allowed states, the $A$ state is incremented, moving up to the next 'layer' in the $z$ direction in Fig. 1. This imposes a hierarchy where the state on subsystem $C$ changes the fastest when we iterate through all global basis states. The subsystem state on $A$ is the leading part and defines horizontal cuts prependicular to the $z$-axis in Fig. 1. Within each layer, fixing the subsystem state $B$ reduces accessible basis states to a *column* which only differ by the $C$ state. Finally, specifying the state on subsystem $C$ fully determines the global state (which is a point) within the layer defined by $A$ and the column additionally defined by $B$.

This structure is advantageous for retrieving the index of a particular state $|\vec{\sigma}_A\rangle \otimes |\vec{\sigma}_B\rangle \otimes |\vec{\sigma}_C\rangle$ with the help of a few lookup tables. Each subsystem contributes an additive part $c_A$, $c_B$, or $c_C$ to the final index, which is then simply given by the sum of these parts. Here, $c_A$ identifies the correct layer, $c_A + c_B$ points to the beginning of the column, and $c_A + c_B + c_C$ yields the final index. In Fig. 1, we discuss the example to retrieve the index of the state $|010\rangle_A \otimes |101\rangle_B \otimes |100\rangle_C$. For this case, the correct layer is the third from the bottom and determined by the state $|\vec{\sigma}\rangle_A = |010\rangle_A$.

The contribution $c_A$ points to the first state (with index 35) in this layer and it is clear that $c_A$ therefore counts the number of all states *prior* to the target layer in the first and second layers in Fig. 1. In Fig. 2, we provide a more detailed view of the same structure by showing each of the eight layers in an individual panel. $c_A = 35$ then corresponds to the first state in the third (target) panel in Fig. 2.

Similarly, we next consider the state on subsystem $B$, $|\vec{\sigma}\rangle_B = |101\rangle_B$, which allows us to skip forward in the global basis to the target column in Fig. 2 where our final state is located. The number of states to skip depends on the number of particles in $B$ and in $A$. The column determined by $|\vec{\sigma}\rangle_A$ and $|\vec{\sigma}\rangle_B$ starts with index $c_A + c_B = 48$. The third panel in Fig. 2 highlights the contribution $c_B = 13$, which can be illustrated as the number of states in the layer occurring before we reach the target state on $B$.

Finally, the state of the $C$ subsystem $|\vec{\sigma}\rangle_C = |100\rangle_C$ determines the location within the column which corresponds to the index of the $C$ state in the $n_C$ sector of the $C$ basis: $c_A + c_B + c_C = 50$ with $c_C = 2$.

## B. General recipe

The idea illustrated in the previous section can be formalized and generalized to any number of subsystems and particles. Similarly to the example from Sec. III A, the global index of a basis state $|\vec{\sigma}\rangle$ is obtained by summing up contributions from all subsystems:

$$\text{index}(|\vec{\sigma}\rangle) = \sum_{k=0}^{N-1} c_k(n_k, \lambda_k, \vec{\sigma}^{(k)}), \qquad (12)$$

Each coefficient $c_k$ is positive and depends on the number of particles $n_k$ in the subsystem $P_k$, the number of particles $\lambda_k$ in the *previous* subsystems $P_0 \ldots P_{k-1}$ and on the subsystem state on $P_k$, $|\vec{\sigma}^{(k)}\rangle$. Hence, the final index monotonously increases while traversing through the subsystems. It corresponds to the cumulative number of global basis states occurring in our chosen canonical order before the subsystem state reaches the target state. By cumulative we mean here that we first count such states to fix the $P_0$ state and *from here* we start counting from zero again to determine the number of global states we have to increment before $P_1$ reaches the target state, i.e. we keep the state on $P_0$ fixed (analogous to first fixing the layer, and then counting the number of states to reach the target column in the three dimensional example).

Therefore, we traverse through all subsystems, beginning with $P_0$, respecting the total particle number constraint of $n$. All allowed states [56] on $P_0$ are sorted with regard to their particle number which we denote $n_0$. For fixed subsystem particle number $n_0$, each subsystem state has a unique, zero based index $\text{index}_0(|\vec{\sigma}\rangle_0)$. An example of such an order is given in Table I. Given a subsystem state $|\vec{\sigma}\rangle_0$ with $n_0$ particles on the first subsystem $P_0$, there are $D_Q(L - L_0, n - n_0)$ possible states in the *complement* of $P_0$ of size $L - L_0$ with $n - n_0$ particles that fulfill the global particle number constraint of $n$. The coefficient $c_{P_0}$ from Eq. (12) counts all states in the global basis which occur before $P_0$ reaches the target state. For each prior subsystem state on $P_0$, we hence have to take into account all configurations on the complement of $P_0$ which can be paired with the $P_0$ state while fulfilling the global constraint of fixed particle number $n$.

Once we have determined $c_0$, the $P_0$ state is fixed and the dimensionality of the problem is effectively reduced from $N$ to $N - 1$ subsystems. Now, the same strategy can be applied to $P_1$. Since we no longer have to worry about $P_0$, to determine $c_1$, we only have to count combinations with legal states in the remaining subsystems $P_2, P_3, \ldots P_{N-1}$ of total size $L - L_0 - L_1$ and the effective particle number constraint is $n - n_0$ since $n_0$ particles are already bound to $P_0$. This recursive scheme is carried out through the entire system until the last subsystem $P_{N-1}$ is reached and the final index is recovered.

Each contribution $c_k$ is composed of two parts: *(i)* an offset counting all basis states with subsystem particle number lower than $n_k$ and *(ii)* a stride determining by how much the global index increases, if the subsystem

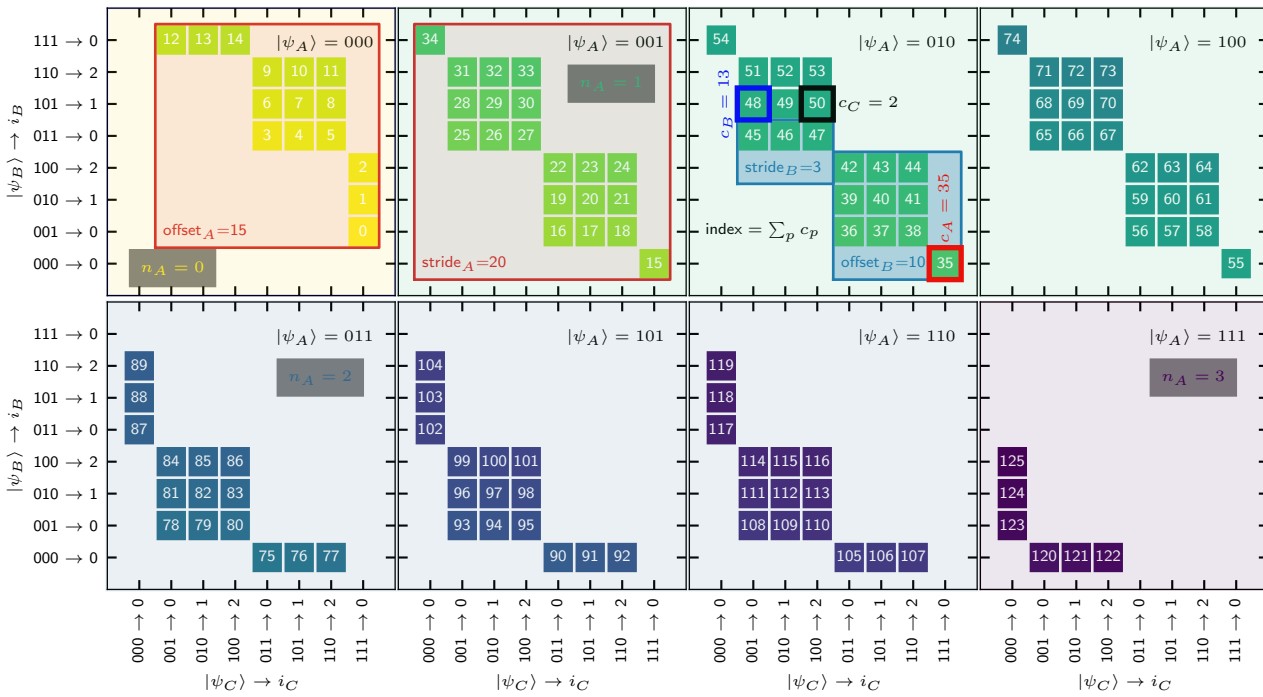

FIG. 2. Indexing of all basis states in the $n = 4$ particles sector on $L = 9$ sites with $Q = 2$ states each based on partitioning of the system into three subsystems $A$, $B$, and $C$. The subsystem basis states are grouped by the subsystem particle number and the indices within each subsystem particle number sector are illustrated by $100 \to 2$ which means that the state $|100\rangle$ has index 2 in the sector where the subsystem has one particle. The different panels correspond to horizontal slices (one for each basis state on the $A$ subsystem in Fig. 1). The emerging block structure in this figure is the key concept behind the algorithm, each block corresponds to fixed particle numbers for all subsystems. Since each subsystem particle number may have a different size, there is a hierarchy of offsets (first index in the global ordering where the subsystem particle number sector begins) and strides (by how much the global index grows if a subsystem state is incremented to the next legal option within the sector).

state is incremented within the particle number sector $n_k$. Together with the zero based index of the subsystem state $|\vec{\sigma}^{(k)}\rangle$ in the subsystem particle number sector, we then have the explicit expression

$$c_k = \text{offset}_k(n_k, \lambda_k) + \text{stride}_k(n_k, \lambda_k) \cdot \text{index}_k(\vec{\sigma}^{(k)}) \quad (13)$$

Here, $n_k$ is the local particle number within the $k$-th subsystem, and $\lambda_k$ is the total particle number contained in the subsystems $P_0$ to $P_{k-1}$ to the left of $P_k$:

$$\lambda_k = \sum_{i=0}^{k-1} n_i. \quad (14)$$

The state of the subsystem is $|\vec{\sigma}^{(k)}\rangle$ and has a zero based index $\text{index}_k(\vec{\sigma}^{(k)})$ in each subsystem particle number sector. Similarly to the two-dimensional search [7], $\text{index}_k(\vec{\sigma}^{(k)})$ refers to a local lookup table of $P_k$ that maps $|\vec{\sigma}^{(k)}\rangle$ to an integer running from zero to $D_Q(L_k, n_k) - 1$. The mapping within a particle number sector $n_k$ can be arbitrary but has to be bijective such that each subsystem state maps to a unique number within the given interval. One possible lookup table for subsystems of length $L_k = 3$ with $Q = 2$ is listed in Table I which is used in the example Fig. 1 and Fig. 2

| $n_A$ | $|\vec{\sigma}_A\rangle$ | $\text{index}_A\vec{\sigma}_A$ | | $n_A$ | $|\vec{\sigma}_A\rangle$ | $\text{index}_A\vec{\sigma}_A$ |
|---|---|---|---|---|---|---|
| 0 | $|000\rangle$ | 0 | | 2 | $|011\rangle$ | 0 |
| 1 | $|001\rangle$ | 0 | | 2 | $|101\rangle$ | 1 |
| 1 | $|010\rangle$ | 1 | | 2 | $|110\rangle$ | 2 |
| 1 | $|100\rangle$ | 2 | | 3 | $|111\rangle$ | 0 |

TABLE I. Example for a lookup table for subsystem $A$ from Fig. 1 and Fig. 2. The choice within each particle number sector can be arbitrary.

### 1. Offset

To derive the expression for the required offsets, we start from the first subsystem $P_0$ and a given state $|\vec{\sigma}^{(0)}\rangle$ with $n_0$ particles. The offset counts all possible states with a lower particle number than $n_0$. Due to the globally fixed particle number $n$, there is lower bound for the number of particles $n_0^{\text{low}}$ that have to be placed in the subsystem $P_0$. If the complement of $P_0$ is large enough to accommodate all $n$ particles,

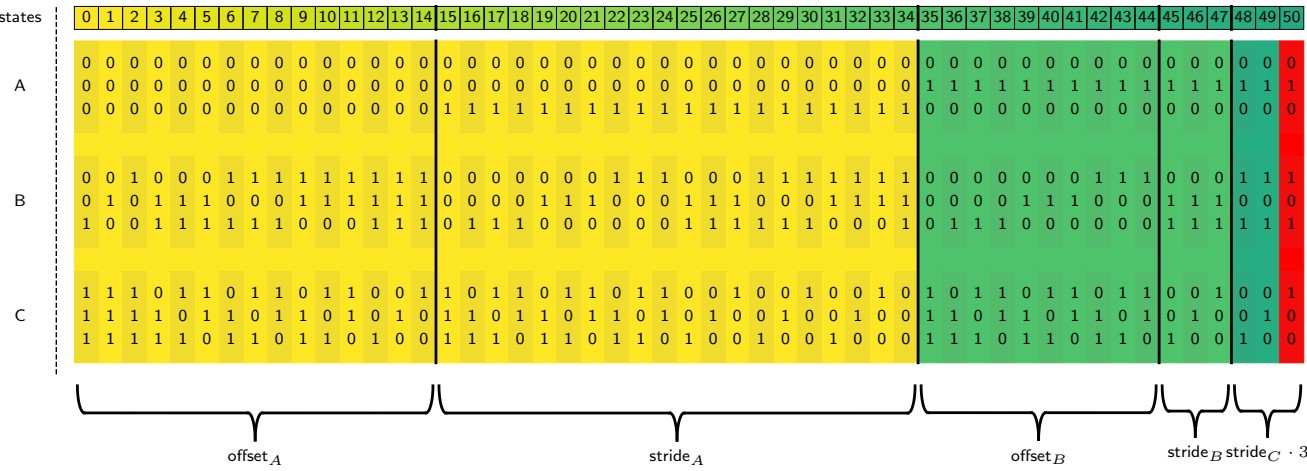

FIG. 3. Example breakdown for finding the index of the 50-th state (red) $|010\rangle_A \otimes |101\rangle_B \otimes |100\rangle_C$ following the example Fig. 1 and Fig. 2. We explicitly show the first 50 states in the basis and highlighted the contribution of the three subsystems $A$, $B$, and $C$ in yellow, light green, and dark green respectively. Beginning with $|010\rangle_A$, we find that it has $n_A = 1$ and we hence skip ahead to the first state with $n_A = 1$, which is state $\text{offset}_A (n_A, \lambda_A) = 15$. Incrementing the state in $A$ in this sector increases the global index by 20, this is $\text{stride}_A (n_A, \lambda_A) = 20$. $|010\rangle_A$ has $\text{index}_A(\vec{\sigma}^{(A)}) = 1$, see Table I, in the subsystem basis of the $n_A = 1$ sector and we hence skip forward to the global index $c_A = \text{offset}_A + \text{index}_A \cdot \text{stride}_A = 35$. We observe that in the slice ahead from global index 35 to our target index 50, the $A$ state no longer changes and we can now move on to subsystem $B$. The state on subsystem $B$ is $|101\rangle_B$, which is located in the $n_B = 2$ sector. We have to skip ahead the global index by $\text{offset}_B (n_B, \lambda_B) = 10$ to reach the first state in the $n_A = 1$, $n_B = 2$ sector. In this sector, we see that the global index increases by $\text{stride}_B (n_B, \lambda_B) = 3$ if we increment the $B$ state. The state $|101\rangle_B$ has $\text{index}_B(\vec{\sigma}^{(B)}) = 1$ in the subsystem basis, again see Table I, of the $n_B = 2$ sector and we hence have to increment the global index by $c_B = \text{offset}_B (n_B, \lambda_B) + \text{stride}_B (n_B, \lambda_B) \cdot \text{index}_B(\vec{\sigma}^{(B)}) = 13$ to reach the first state in the global basis with the correct subsystem states on $A$ and $B$. This state has the index $c_A + c_B = 48$ and fulfills the constraint $|\vec{\sigma}^{(A)}\rangle = |010\rangle$ and $|\vec{\sigma}^{(B)}\rangle = |101\rangle_B$. Since $C$ is the last subsystem, it does not have an offset, $\text{offset}_C (n_C, \lambda_C) = 0$, and its $\text{stride}_C (n_C, \lambda_C) = 1$. Therefore, by incrementing the state on subsystem $C$ directly increments the global basis index by one. Its index is $\text{index}_C(\vec{\sigma}^{(C)}) = 3$ yielding $c_C = 2$ and the final index is retrieved $c_A + c_B + c_C = 50$.

$n_0^{\text{low}} = 0$, else, it has to reflect the fact that at least $n_0^{\text{low}} = \max(0, n - (Q-1)(L - L_0))$ need to be placed in the subsystem $P_0$ to satisfy the constraint. For each valid particle number $k_0$ on $P_0$, there are $D_Q(L_0, k_0)$ possible configurations for states on $P_0$. Each such state can be combined with any state in the *complement* (all other subsystems) of length $L - L_0$ with $n - k_0$ particles in it, and there are $D_Q(L - L_0, n - k_0)$ choices for this. Therefore, we find a total of $D_Q(L_0, k_0)D_Q(L - L_0, n - k_0)$ states with the constraints of $k_0$ particles in $P_0$ and $n$ particles in total. In sum, to account for each *valid* subsystem particle number sector $k_0$ that is lower than $n_0$, we find

$$\text{offset}_0(n_0) = \sum_{k_0=n_0^{\text{low}}}^{n_0-1} D_Q(L_0, k_0)D_Q(L - L_0, n - k_0). \tag{15}$$

For the next subsystem, $P_1$, it is crucial to realize that the state and particle number on $P_0$ is already fixed, effectively reducing the dimensionality of the remaining problem by one. We hence only need to consider the remaining $n - n_0$ particles. The length of the complement

$C_1 = \overline{P_0 \cup P_1}$ is

$$\Gamma_1 = L - L_0 - L_1 \tag{16}$$

and it needs to host $n - n_0 - k_1$ particles, if $P_1$ hosts $k_1$ particles. With the minimal allowed number of particles in $P_1$ given by $n_1^{\text{low}} = \max(0, n - n_0 - (Q-1)\Gamma_1)$, the offset for $P_1$ is given by

$$\text{offset}_1(n_1, \lambda_1) = \sum_{k_1=n_1^{\text{low}}}^{n_1-1} D_Q(L_1, k_1)D_Q(\Gamma_1, n - \lambda_1 - k_1), \tag{17}$$

where $\lambda_1 = n_0$, the number of particles already locked into $P_0$.

Taking the general form for the length of complement $C_i$ of subsystem $P_i$,

$$\Gamma_i = L - \sum_{l=0}^{i-1} L_l, \tag{18}$$

we can generalize the offset for subsystem $P_i$ to

$$\text{offset}_i(n_i, \lambda_i) = \sum_{k_i=n_i^{\text{low}}}^{n_i-1} D_Q(L_i, k_i) D_Q(\Gamma_i, n - \lambda_i - k_i).$$

$$(19)$$

Depending on the particle number and the subsystem length, there might be a minimal amount of $n_i^{\text{low}}$ particles that have to be placed in $P_i$ in order to match the global constraint of $n$ particles. The general form is given by $n_i^{\text{low}} = \max(0, n - \lambda_i - (Q-1)\Gamma_i)$. Importantly, for the last subsystem $P_{N-1}$, since the global number of particles is fixed, its particle number equals the lower bound $n_{N-1}^{\text{low}}$ yielding $\text{offset}_{N-1}(n_{N-1}, \lambda_{N-1}) = 0$.

### 2. Stride

In addition to the offsets, which bring us to the beginning of the relevant subsystem particle number sectors, we need to determine the increase in the global index if the subsystem state is incremented within the particle number sector $n_k$.

To understand the stride, let us again start with the first state $|\vec{\sigma}^{(0)}\rangle$ on $P_0$ with $n_0$ particles. The lookup table for $P_0$ assigns a unique index $i_0 = \text{index}_0(\vec{\sigma}^{(0)})$ to $|\vec{\sigma}^{(0)}\rangle$ which means that $i_0$ subsystem states are ranked lower than $|\vec{\sigma}^{(0)}\rangle$ within the same particle number sector $n_0$. Sec. III E discusses the tables and their construction in more detail. As pointed out in the previous paragraph, the complement $C_0$ of $P_0$ contains all subsystems $P_1$ to $P_{N-1}$ and is of size $\Gamma_0 = L - L_0$. The stride is the number of states in the complement such that the total particle number constraint $n$ is fulfilled. For any state in $P_0$ with $n_0$ particles, this number is

$$\text{stride}_0(n_0, \lambda_0) = D_Q(L - L_0, n - n_0).\qquad (20)$$

Hence, the number of all possible basis states that can be constructed with the $i_0$ subsystem states that are ranked lower than $|\vec{\sigma}^{(0)}\rangle$ is simply $D_Q(L - L_0, n - n_0) \cdot i_0$.

Similarly to the offset, this reduces the dimensionality of the problem when we move to the second subsystem $P_1$. Again, we use its lookup table to obtain the index $i_1 = \text{index}_1(\vec{\sigma}^{(1)})$ of $|\vec{\sigma}^{(1)}\rangle$ with $n_1$ particles. Since $n_0$ particles are a already placed in $P_1$ its complement $C_1$ of size $\Gamma_1 = L - L_0 - L_1$ has to contain $n - n_0 - n_1$ particles leading to ($\lambda_1 = n_0$)

$$\text{stride}_1(n_1, \lambda_1) = D_Q(L - L_0 - L_1, n - n_0 - n_1)\quad (21)$$

possibilities for *each* state with $n_1$ particles in $P_1$. Therefore, the stride contribution, counting all states with the constraint $|\vec{\sigma}^{(0)}\rangle$ on $P_0$ and a lower index than $i_1$ on $P_1$, is $D_Q(L - L_0 - L_1, n - n_0 - n_1) \cdot i_1$.

Following this scheme, we can generalize the stride contribution for the $i$-th subsystem with the state $|\vec{\sigma}^{(i)}\rangle$ and $n_i$ particles. Its index is again retrieved from $P_i$'s lookup table: $i_i = \text{index}_i(\vec{\sigma}^{(i)})$. The previous subsystems $P_0$ to

$P_{i-1}$ contain $\lambda_i = \sum_{i=0}^{i-1} n_i$ particles reducing the global constraint to $n - \lambda_i$ particles on $P_i$ and its complement $C_i$. The general form of stride counting the number of possible states with $n - \lambda_i - n_i$ in the complement $C_i$ of size $\Gamma_i = L - \sum_{i=0}^{i-1} L_i$ is

$$\text{stride}_i(n_i, \lambda_i) = D_Q(\Gamma_i, n - \lambda_i - n_i).\qquad (22)$$

Hence, the number of states with the constraints $|\vec{\sigma}^{(i)}\rangle$ on $P_i$ for $i = 0, \ldots, i-1$ and lower ranked subsystem states on $P_i$ with $n_i$ particles is $D_Q(\Gamma_i, n - \lambda_i - n_i) \cdot i_i$. Since the last subsystem does not have a complement, its stride is simply one: $\text{stride}_N(n_N, \lambda_N) = 1$.

We have transformed the three-dimensional example from Fig. 1 into a list shown in Fig. 3 which highlights the individual contributions in form of offsets and strides. A detailed explanation is given in the caption.

### C. Two important limits

Next we want to discuss two important limits of the algorithm: $N = 2$ and $N = L$. We start by discussing the original approach by Lin [7] that is based on two subsystems. Then, we illustrate the opposite limit [50] which consists of $N = L$ subsystems of size one.

#### 1. Two subsystems ($N = 2$)

The case with two subsystems is special as a state in $P_0$ with $n_0$ particles fixes the number of particles in $P_1$ due to the global constraint: $n_1 = n - n_0$. In this case, we can store the individual contributions $c_0$ and $c_1$ directly into two lookup tables that label the local basis states as shown in Table I. Since $P_1$ is the last subsystem we find that the offset is zero and the stride is always one. Hence, $c_1$ reduces to $\text{index}_1(\vec{\sigma}^{(1)})$ which is simply the bare lookup table we discussed before. This corresponds to system $A$ with $J_a(I_a)$ in table II of Ref. [7]. Note that Ref. [7] does *not* work with zero-based indexing which is used throughout this manuscript and the accompanying code.

Now, to incorporate the contribution of the first subsystem $P_0$, we overwrite its original lookup table – which maps $|\vec{\sigma}^{(0)}\rangle$ to a unique index $\text{index}_0(\vec{\sigma}^{(0)})$ – simply by its total contribution:

$$c_0 = \text{offset}_0(n_0, \lambda_0) + \text{stride}_0(n_0, \lambda_0)\,\text{index}_0(\vec{\sigma}^{(0)})\quad (23)$$

The offset and stride are given by Eq. (15) and Eq. (20):

$$\text{offset}_0(n_0, \lambda_0) = \sum_{k_0=n_0^{\text{low}}}^{n_0-1} D_Q(L_0, k_0) D_Q(L_1, n - k_0)$$

$$\text{stride}_0(n_0, \lambda_0) = D_Q(L_1, n - n_0)$$

The newly overwritten table corresponds to part $B$ with $J_b(I_b)$ in the table II from Ref. [7].

While the trick to store the coefficients directly into the lookup table works for $N = 2$ due to the global constraint, the scheme is not possible for $N > 2$ and we have to account for this by tracking the particle number using $\lambda_i$.

### 2. $L$ subsystems ($N = L$)

The opposite limit, evaluating $N = L$ subsystems of size one can be done "on-the-fly" as it does not require the use of lookup tables. Since each system is of size one, it can have at most $Q$ different states $|q_i\rangle$ with $q_i = 0, \ldots, Q - 1$. Therefore, there is only one state in each particle number sector in $P_i$ inducing $\text{index}_i(\vec{\sigma}^{(i)}) = 0$. Then, the contribution of the $i$-th subsystem simplifies to:

$$c_i = \text{offset}_i(q_i, \lambda_i) = \sum_{k_i = n_i^{\text{low}}}^{q_i - 1} D_Q(\Gamma_i, n - \lambda_i - k_i) \quad (24)$$

We have outlined the algorithm for $N = L$ in Algo. 1 and refer it as the "on-the-fly" implementation throughout the rest of the manuscript.

In the binary case ($Q = 2$), the formula to compute the index was already derived in Ref. [48, 49] and applied to physical problems by Ref. [50].

---

**Algorithm 1:** On-the-fly

**Data:** $|\vec{\sigma}\rangle = |q_0; \ldots; q_{L-1}\rangle$
index, $\lambda = 0$        /* initializing variables */
$\Gamma = L - 1$
**for** $0 \leq i < L - 1$ **do**
     **for** $0 \leq k < q_i$ **do**
         **if** $n - \lambda - 1 - k \leq (Q-1)\Gamma$ **then**
            |   index = index + $D_Q(\Gamma, n - \lambda)$
         **end**
         $\lambda = \lambda + 1$
     **end**
     $\Gamma = \Gamma - 1$
**end**
**return** index;

---

### D. Enumerative encoding

The presented enumeration of basis states is an old problem in computer science and combinatorics [57]. In particular, Cover presented a generic ansatz in 1973 to compute the lexicographic one-to-one mapping and its inverse [49]. The idea behind his approach reflects the divide-and-conquer ansatz used in the derivation of our multidimensional search algorithm. In fact, we can use his formulation to derive our algorithm.

To formulate the problem in a computer science language, let $\vec{x} = (x_0, \ldots, x_{N-1})$ be a *word* of length $N$ and $x_i \in \{0, \ldots, Q - 1\}$ the *letters* from an alphabet of size $Q$. Then, the lexicographic order, $\vec{x} < \vec{y}$, is defined by $x_i < y_i$ where $i$ is the smallest index with $x_i \neq y_i$.

Given any *arbitrary* subset $\mathcal{S}$ of all possible words of length $N$, we can use Cover's formula given in proposition 2 in Ref. [49] to find the lexicographic one-to-one mapping: $\mathcal{S} \to \{0, \ldots, |S| - 1\}$. There, he defines the number of elements in $\mathcal{S}$ for which the first $k$ letters are $(x_0, \ldots, x_k)$ by $n_{\mathcal{S}}(x_0, \ldots, x_k)$. The general formula that provides the desired mapping for $\vec{x}$ is:

$$\text{index}(\vec{x}) = \sum_{k=0}^{N-1} \sum_{l=0}^{x_k - 1} n_{\mathcal{S}}(x_0, \ldots, x_{k-1}, l) \quad (25)$$

To demonstrate the generality of this ansatz, we have chosen a generic – not number conserving – set:

$$\mathcal{S} = \{(0,2,0), (0,2,1), (1,0,1), (2,0,0), (2,2,0), (2,2,1)\}.$$

The set is already lexicographically ordered and we can illustrate the counting of $n_{\mathcal{S}}$. For example, the number of elements starting with $(1)$ is $n_{\mathcal{S}}(1) = 1$ and with $(0,2)$ is $n_{\mathcal{S}}(0,2) = 2$. Following the Eq. (25), we derive the index of the last element $\vec{x} = (2, 2, 1)$ which is 5:

$$\begin{aligned}
&\text{index}(\vec{x}) \\
&= \underbrace{n_{\mathcal{S}}(0) + n_{\mathcal{S}}(1)}_{k=0} + \underbrace{n_{\mathcal{S}}(2,0) + n_{\mathcal{S}}(2,1)}_{k=1} + \underbrace{n_{\mathcal{S}}(2,2,0)}_{k=2} \\
&= 2 + 1 + 1 + 0 + 1 = 5
\end{aligned}$$

Similarly to our multi-dimensional search algorithm, the first contribution, $k = 0$, takes care of all elements in $\mathcal{S}$ that have a smaller letter than $x_0$. This refers to the first contribution $c_A$ in Fig. 1 that identifies the correct plane. The second part, $k = 1$, refers to $c_B$ and jumps to the correct column. Lastly, $k = 2$ takes care of the last part and refers to the contribution $c_C$.

To relate this ansatz to our number constraint, we first use Eq. (25) to derive the $N = L$ limit with arbitrary $Q$. The contribution of the $k$-th subsystem is

$$c_k = \sum_{l_k=0}^{x_k - 1} n_{\mathcal{S}}(x_0, \ldots, x_{k-1}, l_k). \quad (26)$$

The number of possible configurations in $\mathcal{S}$ that begin with $(x_0, \ldots, x_{k-1})$ and fulfill the particle number constraint $n = \sum_k x_k$ are

$$n_{\mathcal{S}}(x_0, \ldots, x_{k-1}, l_k) = D_Q(L - 1 - k, n - \lambda_k - l_k).$$

$L - 1 - k$ is the length of the complement defined earlier by $\Gamma_k$. $\lambda_k$ is the number of particles contained up to subsystem $P_k$: $\lambda_k = \sum_{s=0}^{k-1} x_s$. As we discussed in the preceding section, there might be a constraint on $l_k$ restricting the sum in Eq. (26) to $l_k \in \{n_k^{\text{low}}, \ldots, x_k - 1\}$. This can be extracted from the definition of $n_{\mathcal{S}}(\ldots)$ which is simply zero if $l_k < n_k^{\text{low}}$. We have derived the same contribution for $N = L$ given in Eq. (24) using the general formalism from Cover.

Similarly, we can derive the offset and stride for the generic case. For simplicity, we choose an equal partitioning where all subsystem sizes are identical. In this case, the alphabet is growing exponentially with system size and each subsystem can have $M = 2^{L/N}$ states. Therefore, each $x_i = 0, \ldots, M-1$ can take exponentially many values. To define a lexicographical order, we first have to impose a canonical ordering within each subsystem. Following the previous section, all $M$ states are ordered by particle their number (lower number first) and we use a lookup table, *cf.* Table I, to impose the order within each particle number sector. Each letter $x_i$ refers to a substate on $P_k$ and has an associated particle number $n(x_i) = 0, \ldots, L/N(Q-1)$. The subset $\mathcal{S}$ is defined by the global particle number constraint $n$ and we use Eq. (25) to derive the index of $\vec{x} \in \mathcal{S}$. The contribution of the $k$-th subsystem is:

$$c_k = \sum_{l_k=0}^{x_k-1} n_{\mathcal{S}}(x_1, \ldots, x_{k-1}, l_k) \qquad (27)$$

Note that the sum runs over exponentially many letters. To avoid adding this exponential overhead, we simply group letter $l_k \in \{0, \ldots, x_k - 1\}$ into particle number sectors $m_k$ on $P_k$:

$$\sum_{l_k=0}^{x_k-1} \to \sum_{m_k=0}^{n(x_k)} \sum_{l_k=0}^{x_k-1} \delta_{m_k, n(l_k)}$$

$n(l_k)$ is the number of particles of the $l_k$-th state on $P_k$. For a given particle number $m_k < n(x_k)$, the number of states contained in the sum are $D_Q(L/N, m_k)$. Crucially, note that the number of words in $\mathcal{S}$ starting with $(x_0, \ldots, x_{k-1}, l_k)$ only depends on number of particles contained in subsystem $P_0$ to $P_k$: $\lambda_k + n(l_k)$. Therefore, grouping the states according to their particle number greatly simplifies the equation as we can replace $n_{\mathcal{S}}(x_1, \ldots, x_{k-1}, l_k)$ by $n_{\mathcal{S}}(\lambda_k, m_k)$:

$$c_k = \sum_{m_k=0}^{n(x_k)-1} D_Q(L/N, m_k) n_{\mathcal{S}}(\lambda_k, m_k) .$$
$$+ \text{index}_k(\vec{\sigma}^{(k)}) n_{\mathcal{S}}(\lambda_k, n(x_k))$$

Here, $|\vec{\sigma}^{(k)}\rangle$ refers to the state associated with the letter $x_k$ and $\text{index}_k(\vec{\sigma}^{(k)})$ is the index in the particle number sector $n_k = n(x_k)$. This form makes the origin of the offset and stride clear. To finally determine $n_{\mathcal{S}}(\lambda_k, m_k)$, we can use the same argument as in the previous section. Given the total constraint $n$, we already have $n - \lambda_k - m_k$ particles distributed on subsystems $P_0$ to $P_k$. Therefore, the number of possible configurations in $\mathcal{S}$ starting with any string $(x_0, \ldots, x_k)$ that contains $\lambda_k + m_k$ particles is $n_{\mathcal{S}}(\lambda_k, m_k) = D_Q(\Gamma_k, n - \lambda_k - m_k)$ where $\Gamma_k$ is the length of the complement. Note that Cover's formula implicitly includes the lower bound on the particles on $P_k$ as $n_{\mathcal{S}}(\lambda_k, m_k) = 0$ of $m_k < n_k^{\text{low}}$. Hence, we have derived our expression for $c_k$ from Eq. (13) with the same offsets Eq. (19) and strides Eq. (22).

### E. Lookup tables

An efficient implementation of our multidimensional search algorithm uses two kinds of lookup tables. One is used to store the offsets and strides that are computed with Eq. (19) and Eq. (22). Both, the offsets and strides, depend on $n_i$ and $\lambda_i$ that can not be greater than $n \leq (Q-1)L$. Therefore, the memory required to store all possible coefficients for all $N$ subsystems is smaller than $Nn^2$ and fits easily on any hardware.

However, the size of the other type of lookup table scales exponentially with the subsystem size and reducing its volume is the motivation behind our work by introducing more subsystems. To recall, each subsystem has a lookup table that defines a canonical order within $P_i$ ignoring the rest of system: For each subsystem particle number sector $n_i$, the table provides a *one-to-one* mapping between subsystem states $|\vec{\sigma}^{(i)}\rangle$ and an index, $\text{index}_i(\vec{\sigma}^{(i)})$, from zero to $D_Q(L_i, n_i) - 1$. Note that the system $P_i$ has different particle number sector where each has its own zero-based labeling. An example is shown Table I. The index, together with the offset and stride, defines the subsystem contribution $c_i$.

The size of the lookup table $\text{index}_i(\vec{\sigma}^{(i)})$ scales exponentially with the subsystem size: $Q^{L_i}$. While the overhead coming from this table is manageable and does not hamper performance for $L_i \sim 10$, it quickly becomes a bottleneck for matrix-free applications in large eigenvalue problems. Therefore, in order to break the exponential increase, the system is split into multiple parts keeping the individual subsystem sizes small. Splitting the system into $N = 2$ parts, as proposed by Ref. [7], helps to delay the problem but it is an unsatisfying approach for $L \gtrsim 30$. In these cases, partitioning the system in more than two subsystems is required to reduce the memory overhead.

#### 1. Implementation

We have implemented and tested three approaches to encode the lookup table $\text{index}_i(\vec{\sigma}^{(i)})$:

(i) memory-aligned list

(ii) lexicographical order in a tree-based associative map

(iii) combinatorial *on-the-fly*

The first option uses memory-aligned indices that are accessed using the integer representation of the state $|\vec{\sigma}^{(i)}\rangle$ similar to Ref. [7]. For example, we require two bits to encode a single state $|q\rangle$ for $q = 0, \ldots, 3$ with $Q = 4$. We denote the number of bits necessary to store a single state by $\text{Nbits}_Q = \text{ceil}(\log(Q)/\log(2))$. The state $|3; 2; 1; 0\rangle$ with $L = 4$ spans over eight bits: (11100100) where two consecutive bits refer to a single qudit state. The bit string encodes the integer 228 and we, therefore, store the index of the state $|3; 2; 1; 0\rangle$ at the 228-th position.

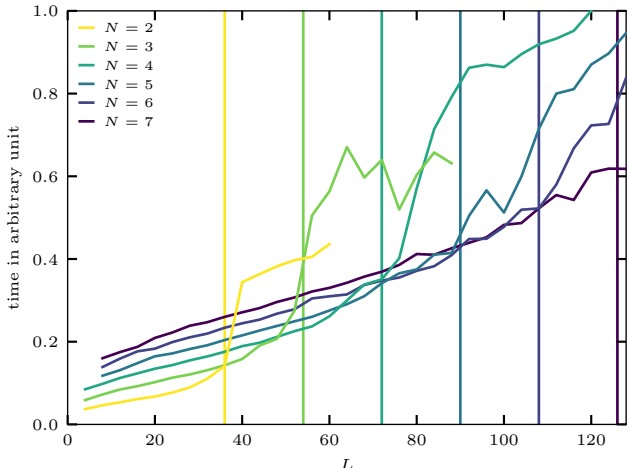

FIG. 4. Runtime in arbitrary units to enumerate randomly generated trial states for a system of size $L$ with $Q = 2$. $N$ refers to the number of subsystems that are distributed "most" equally. The vertical lines mark the system size where the memory of the lookup table with size $2^{\texttt{ceil}(L/N)}$ exceeds the cache, here 4MB, of the processor. All computations were done using a two unsigned integer with 64 bits each for state representation.

The second implementation (ii) might be advantageous in the limit of small filling fractions $n \ll L$. A table encoding a system of size $L$ using the (i) requires $2^{\texttt{Nbits}_Q \cdot L}$ entries. However, in the limit of small fillings, most entries will never be used. By using a lexicographical order that only includes the valid states, the subsystem length can be chosen significantly bigger than in the first case.

Lastly, we can simply exploit Algo. 1 to compute the indices on-the-fly without actually storing the subsystem states. We actually use the algorithm to assign the unique indices to the subsystem states when tables in form (i) and (ii) are constructed. Again, the precise order is arbitrary as long as the mapping is one-to-one.

### F. Sparse tensor storage

A core module of the DanceQ library is the `Operator` class, which provides an easy interface to handle arbitrary tensor products defined on a system consisting of $L$ sites with a local Hilbert space dimension $Q$. Besides handling and organizing any input, it allows for a highly optimized on-the-fly matrix-vector multiplication without storing the exponentially large matrix. For optimal performance, the `Operator` class employs a similar divide-and-conquer approach. This involves merging several local terms that act on the same sites, a strategy that enhances efficiency and reduces computational complexity. In particular, we identify subclusters of $N_{\text{tensor}}$ sites of the system and merge all local operators fully supported in this subcluster into a single sparse matrix

of size $Q^{N_{\text{tensor}}}$.

Consider for example a one-dimensional spin chain of length $L = 30$ (we use periodic boundary conditions where we identify site 30 refers site 0):

$$H = \sum_{i=0}^{29} S_i^x S_{i+1}^x + S_i^y S_{i+1}^y + S_i^z S_{i+1}^z + \sum_{i=0}^{29} S_i^z \quad (28)$$

To apply the Hamiltonian to a product state, we have execute all $4 \cdot 30$ local operators. In order to reduce this complexity that scales with $L$, we assign three *overlapping* subclusters of size $N_{\text{tensor}} = 11$:

$$\mathcal{C}_0 = \{0, \ldots, 10\}, \mathcal{C}_1 = \{10, \ldots, 20\}, \mathcal{C}_2 = \{0, 20, \ldots, 29\}$$

Note that the subclusters need to overlap to encompass all terms. This allows us only to store three sparse matrices $\mathcal{S}_i$ of size $2^{11}$, each containing all operators fully supported on the individual clusters $C_i$. For example, all terms that act solely on $\mathcal{C}_0$,

$$H_{\mathcal{C}_0} = \sum_{i=0}^{9} S_i^x S_{i+1}^x + S_i^y S_{i+1}^y + S_i^z S_{i+1}^z + \sum_{i=0}^{10} S_i^z \quad (29)$$

are compressed into $\mathcal{S}_0$. Thus, applying the large tensor matrix reduces the complexity of iterating over $4 \cdot 30$ local operators to only three operators resulting in less state manipulations and computational overhead. Despite the enhanced dimension of the matrices $\mathcal{S}_i$ compared to the two-body terms in Eq. (28), it is bounded by $N_{\text{tensor}}$ and can be chosen such it easily fits in the cache of the processor. We have chosen the default such that the dimension does not exceed $Q^{N_{\text{tensor}}} = 2048$.

Now, given an input state $\psi$, we can extract the corresponding `columnindex` of the sparse matrix for a given cluster $\mathcal{C}_i$ by:

$$\texttt{columnindex} = \sum_{k=0}^{|\mathcal{C}_k|-1} Q^k \psi[k] \quad (30)$$

In our implementation, this index points directly to the memory-aligned coefficients and elements of $\mathcal{S}_i$. To further enhance the computation, the class works with statemasks which are stored within the sparse matrix. Hence, instead of storing the sparse matrix of size $Q^{N_{\text{tensor}}} \times Q^{N_{\text{tensor}}}$, we directly store each *column* of this matrix as a *sparse vector*.

While the above description refers to only nearest-neighbor operators acting on two sites, its generalization is straightforward and implemented in the class. Note that the choice of subclusters does not correspond the partition of our multidimensional search algorithm.

To apply a column of the cluster-local operator to an element of the input vector (with a corresponding basis state), we effectively iterate over all configurations on the complement of the cluster for each nonzero element of the sparse matrix to calculate the contributions to the result vector. The bookkeeping in the innermost loop is performed using cheap bitwise logical operations.

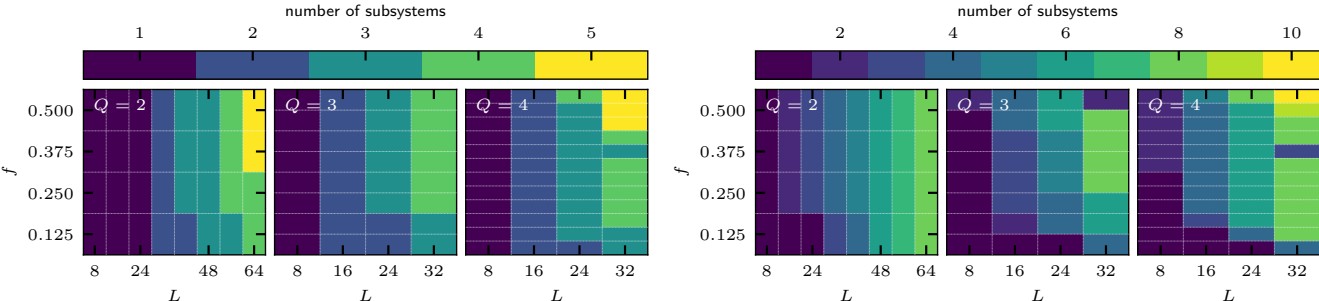

FIG. 5. Optimal number of subsystems for two different implementation of the lookup tables using the memory aligned list (left) and a lexicographical order (right). To determine the optimal $N_Q^{\mathrm{opt}}$, we have chosen the most equal partition and measured the time it takes to retrieve indices of randomly generated trial states. The optimal $N_Q^{\mathrm{opt}}$ has the lowest runtime. $L$ refers to the total system size and $Q$ to the local Hilbert space dimension. The filling fraction is defined by number of particles in the total system divided by the maximal number of particles possible: $f = \frac{n}{(Q-1)L}$. We have not shown a computation for the on-the-fly approach (iii) as the optimal number subsystems is simply $N = L$ for all cases. We further find that option (i) is superior for all filling fractions considered here. All computations were done using a single unsigned integer with 64 bits for state representation.

## IV.   PERFORMANCE

For a given length and filling fraction, the performance of the algorithm depends on the number of subsystems and their partitioning. To find the optimal choice, we randomly generate a fixed number of trial states and benchmark the time it takes to retrieve their basis index following the general recipe implemented in [43]. The number of states is of order $10^6$. We used an *Intel i7-7500U* (2.70 GHz) processor with a cache size of 4 MB for these benchmarks.

As a first observation, we find that the performance drops significantly when the memory of the lookup tables exceeds the L3 cache of the processor. This is demonstrated in Fig. 4 which shows the runtime versus system size for different $N$. The vertical lines mark the point where the table exceeds the cache size. Hence, for an optimal performance, the required memory should not exceed this limit.

Given the number of subsystems $N$, the partitioning with the lowest memory usage is the one that divides the whole system into the "most" equal parts. In this case, at most two different subsystem sizes $L_i$ are present: $\mathtt{ceil}(L/N)$ and $\mathtt{floor}(L/N)$. This comes with another advantage as we can use the same lookup tables for all subsystems of equal length, reducing the memory consumption further. Hence, we only consider the most equal partitioning of the system for the rest of the manuscript.

We find that the first option (i), storing the indices in an array which is aligned in memory, is in almost all cases the best choice. This is also true for dilute systems containing only a few particles. Fig. 5 displays the optimal number of subsystems for the first two options using the uniform partition. By an optimal number of subsystems $N_Q^{\mathrm{opt}}$, we mean that an equally sized parti-

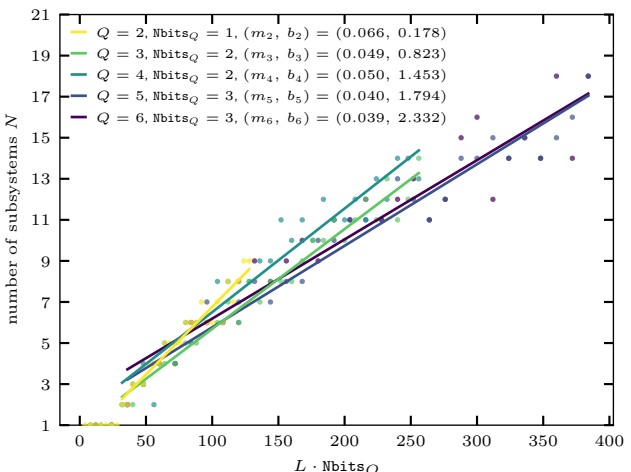

FIG. 6. Optimal number of subsystems for different $Q$ at half filling $(n = L(Q-1)/2)$ versus system size $L\mathtt{Nbits}_Q$. For each system size, we use our test setup with randomly generated trial states and identified optimal system size which is plotted on the $y$-axis. We find a linear scaling and fit $N_Q^{\mathrm{opt}} = m_Q(L \cdot \mathtt{Nbits}_Q + b_Q$ to extract the optimal scaling. $\mathtt{Nbits}_Q = \mathtt{ceil}(\log(Q)/\log(2))$ refers to number required to encode a single site with local Hilbert space dimension $Q$.

tion with $N = N_Q^{\mathrm{opt}}$ has the lowest runtime for our randomly generated test setup. The left panel refers to the memory aligned list (i) and the right panel refers to the lexicographical order (ii). We have evaluated the optimal number of subsystems in both cases for fixed filling fraction $f = \frac{n}{(Q-1)L}$ and length. In both cases we see a clear trend that larger systems and larger filling fractions require more subsystems for an ideal performance.

To understand the scaling of the optimal $N_Q^{\mathrm{opt}}$ for $Q$, we look into the most prominent case at half filling us-

ing (i). Fig. 6 displays the $N_Q^{\text{opt}}$ versus $L \cdot \text{Nbits}_Q$. To recall, $\text{Nbits}_Q = \text{ceil}(\log(Q)/\log(2))$ is the number of bits required to encode a single state of dimension $Q$. We find that the optimal number of subsystems scales linearly with $L \cdot \text{Nbits}_Q$:

$$N_Q^{\text{opt}} = \text{ceil}\left(m_Q(L \cdot \text{Nbits}_Q) + b_Q\right) \qquad (31)$$

We find good agreement for different values of $Q$ and $m_Q \sim 0.05$. This can be understood as an optimal subsystem length $L/N_Q^{\text{opt}}$ such that the table can be stored in the cache:

$$2^{\text{Nbits}_Q \frac{L}{N_Q^{\text{opt}}}} \approx 2^{1/m_Q} \text{ for } b \ll m_Q(L \cdot \text{Nbits}_Q) \qquad (32)$$

To summarize, we recommend to use the most equal partition such that at most two tables have to be stored. Eq. (31) can be used to determine the optimal number of subsystems. However, in practice, the length should be chosen such that the lookup table footprint is smaller since other data needs to be stored in the L3 cache as well.

### A.  Matrix-free multiplication

Many algorithms in computational quantum many-body physics rely solely on matrix-vector multiplications to build for example a Krylov subspace which can used to perform time evolution or to compute ground states and excitations (e.g. by deflation techniques [55]), or other eigenvectors using spectral transformations [58–60]. Krylov space methods are particularly powerful and frequently applied to many physical problems due to the sparseness of the Hamiltonian as it reduces the complexity from a *cubic* for a full diagonalization to an often *linear* scaling with the Hilbert space dimension (which itself remains of course exponential in $L$).

The bottleneck for exact methods is usually the memory requirement to store the sparse Hamiltonian matrix, which scales with the Hilbert space dimension times the number of offdiagonal matrix elements per row (which is typically of order $L$ in the case of nearest-neighbor interactions) for sparse matrix-vector multiplication. Therefore, to reduce the memory further, state-of-the-art computations [30–32, 34, 61] do not store this matrix and instead compute the action of its elements on the input vector on the fly in a massively parallel way. However, to ensure fast computations, each worker process has to have knowledge of the basis states and their associated indices. This is the main contribution of our algorithm as it allows a memory efficient way to perform this type of bookkeeping.

To understand the scaling of the subsystem size within the full matrix-free multiplication [43], we monitored the time it take to perform one such matrix-vector operation. While the performance depended crucially on $N$ and the available cache in the last subsection where we only focused on the lookup, we do not observe this behavior in this case. In fact, we find that the time depends only slightly on the number of subsystems and the best performance was achieved by using a single "subsystem" of size $L$ ($N = 1$) – if it fits in the RAM. We interpret this finding to indicate strong cache interference between data required for the actual multiplications of matrix elements on the vector and data for lookup tables which means that the lookup and the retrieval of the indices plays only a secondary role during the full matrix-vector multiplication and other operations that take place have to be considered. For example, the output wave function (which is usually filling up the whole RAM) is constantly edited and states have to be incremented and manipulated throughout the process. Therefore, to obtain the best performance, we recommend to choose $N$ small, but without consuming any meaningful memory. In other words, the memory footprint of each worker process should be the guiding principle to choose $N$, since the computing time in real-world applications only depends weakly on $N$.

The memory-core ratio is of the order of $4\,\text{GiB}$ on modern platforms and we will use a $4\,\text{GB}$ limit per core as an example for the following discussion. Since memory is the constraining part for Krylov space techniques we do not want to block any significant amount of it. However, storing the lookup table for a single subsystem $N = 1$ blocks the available memory which should be used by the wave function and is quickly exhausted ($L = 27$ for $Q = 2$). In Fig. 7, we show the memory required by the lookup table. In MPI-based programs in this scenario, each worker process is in charge of $4\,\text{GiB}$ and has to store its own table. Therefore, it is not possible that the lookup table takes more than $4\,\text{GiB}$. This is indicated by the red-shaded area where the more subsystems are required to reduce the memory consumption. Blue (yellow) color refer to large (exponentially small) fractions of the $4\,\text{GiB}$ limits used by the table. The default setting of our code and our recommendation is $512\,\text{kiB}$ which refers to a subsystem length of $L_i = 16$ for $Q = 2$ [43]:

$$N_Q^{\text{opt}} = \text{ceil}\left(\frac{\text{Nbits}_Q \cdot L}{16}\right) \qquad (33)$$

This choice is also in agreement with Eq. (31) ($b_Q = 0$) and the linear scaling from Fig. 6 with $m_2 \sim 1/16$. Note that at most two lookup tables are required for the most equal partition.

### V.  CONCLUSION

We presented an elegant solution to efficiently deal with number conserving systems in very large scale, massively parallel calculations where the available memory per core limits space available for lookup tables to map basis states to their index. While an on-the-fly algorithm Algo. 1 exists as the extreme limit with negligible memory requirements, it is the slowest solution. The

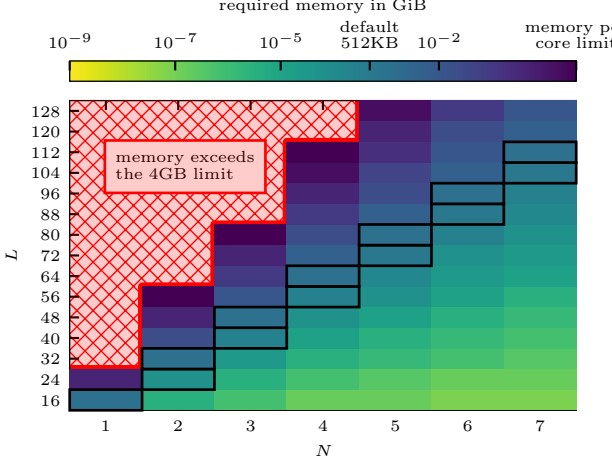

FIG. 7. The figure displays the fraction of memory used to store the largest lookup table to the available memory per processor which we set to 4 GiB here for different system sizes $L$ and number of equally sized subsystems $N$ ($Q = 2$). Blue (yellow) color indicates that a large (exponentially small) portion is used by the lookup table. The red-shaded area indicates systems sizes that require a more subsystems in order to fit the table within the memory of processor. The bottleneck of exact diagonalization is usually memory and the fraction of memory associated to the table should be chosen rather small. For each system size $L$, we have marked the optimal number of subsystems with a black box where the memory required by the lookup table does not exceed our default setting of 512 kiB [43]. Note that the memory consumption of the table is independent of the particle sector.

traditional approach [7] using two subsystems is much faster but requires too much memory for system sizes coming within reach on exascale machines. Our general divide-and-conquer algorithm interpolates between these two limits and provides an optimal balance between computational cost and available memory to overcome these limitations.

We have implemented this algorithm in a general, state-of-the-art, and header-only `C++20` library available at Ref. [43]. The code is user-friendly and allows to exploit the full power of large scale computing facilities making ground-state searches and time evolution for large systems possible. By combining several MPI-threads, our implementation is capable of computing ground states for systems containing 46 spins ($Q = 2$) at half filling. The required memory to store the necessary two wave functions is about 120 TiB.

While the focus of this paper and the accompanying code is on quantum magnetism, it is applicable to other problems of many fermions or bosons with conserved total particle number. The problem to efficiently enumerate states or sequences in lexicographical order extends beyond physics and is important in various areas of computer science [48, 49].

We note in closing that our method is formulated for $L$ identical qudits with $Q$ states per site. At the expense of additional bookkeeping, it is straightforward to generalize our approach to different $Q$ for each site, which is relevant for systems of mixed spin $S$ or for example bose-fermi mixtures [52].

## ACKNOWLEDGMENTS

This work was financially supported by the Deutsche Forschungsgemeinschaft through the cluster of excellence ML4Q (EXC 2004, project-id 390534769). DJL acknowledges support from the QuantERA II Programme that has received funding from the European Union's Horizon 2020 research innovation programme (GA 101017733), and from the Deutsche Forschungsgemeinschaft through the project DQUANT (project-id 499347025). We further acknowledge support by the Deutsche Forschungsgemeinschaft through CRC 1639 NuMeriQS (project-id 511713970). RS acknowledges the AFOSR Grant No. FA9550-20-1-0235.

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

## Appendix A: Hilbert space dimension

We consider a tensor product Hilbert space of local $Q$-dimensional spaces, subject to the constraint that the sum of local excitations $n$ is fixed.

For $Q = 2$, the Hilbert space dimension of a sector $n = 0, \cdots, L$ can be derived combinatorially and is well known to be determined by the binomial coefficient:

$$D_2(L, n) = \binom{L}{n} \tag{A1}$$

However, determining the dimension of each sector for larger local dimension $Q$ is more involved. It is related to the probability of scoring a fixed sum in the throw of $L$ dices with $Q$ faces, *cf.* p. 284, problem 18 in Ref. [62]. In the context of Hilbert space dimensions one of the early applications can be found in Refs. [51, 54].

Here, we provide an elementary derivation of this closed form equation. We approach this problem by defining an equal superposition of all possible computational states

$$|\Psi\rangle = \bigotimes_{i=1}^{L} \left( \sum_{i=0}^{Q-1} |i\rangle \right). \tag{A2}$$

Now, to determine the dimension of a sector with a certain magnetization $n$ we need to identify all states exhibiting the correct magnetization. This problem is equivalent to determining the coefficient of $x^n$ of the polynomial $f(x) = \left(1 + x + \cdots + x^{Q-1}\right)^L$:

$$D_Q(L, n) = \mathrm{coef}_{x^n} \left[ \left(1 + x + \cdots + x^{Q-1}\right)^L \right] \tag{A3}$$

Here we identified the state $|k\rangle_1$ with $x^k$. Each computational state exhibiting the correct magnetization contributes to the coefficient of $x^n$.

We evaluate the polynomial using the finite geometric sum

$$f(x) = \left( \sum_{i=0}^{Q-1} x^i \right)^L = \left( \frac{x^Q - 1}{x - 1} \right)^L = \frac{\left(x^Q - 1\right)^L}{(x - 1)^L} \tag{A4}$$

Then, the denominator is expanded using its Taylor series around $x = 0$:

$$(x - 1)^{-L} = (-1)^{-L} \sum_{k=0}^{\infty} \frac{1}{k!} \left[ \prod_{s=0}^{k-1} (L + s) \right] x^k = (-1)^{-L} \sum_{k=0}^{\infty} \binom{L - 1 + k}{L - 1} x^k \tag{A5}$$

and the nominator is evaluated using the binomial coefficients:

$$\left(x^Q - 1\right)^L = \sum_{k=0}^{L} \binom{L}{k} x^{qk} (-1)^{L-k}. \tag{A6}$$

To obtain the dimension of the sector, Eq. (A5) and Eq. (A6) are multiplied and we evaluate the coefficient of $x^n$:

$$D_Q(L, n) = \sum_{k=0}^{\lfloor n/q \rfloor} (-1)^k \binom{L}{k} \binom{L - 1 + n - qk}{L - 1} \tag{A7}$$

$\lfloor \, \rfloor$ is the lower Gauss bracket.

## Appendix B: Pseudo code

This section presents the pseudo code of the most important functions (i), (ii), and (iii) from Sec. II B. Our DanceQ library initiates the lookup tables that provide the index within a particle number sector, and all necessary offsets and strides from Eq. (19) and Eq. (22). The following functions are implemented by an underlying State class:

- `get_n`$(|\vec{\sigma}\rangle, k)$:
  Returns the number of particles in the subsystem $P_k$.

- `get_minimal_state`$(l, n)$:
  Returns the state with index 0 for a system with $l$ sites and $n$ particles. It has to be consistent with the lookup tables.

- `is_maximal`$(|\vec{\sigma}\rangle, k)$:
  Returns *True* if the subsystem state on $P_k$ is the last state for its particle number sector in $P$. It has to be consistent with the lookup tables.

- `increment_local`$(|\vec{\sigma}\rangle, k)$:
  Returns the next state within the same particle number sector of $|\vec{\sigma}^{(k)}\rangle$ on subsystem $P_k$ according to the lookup table.

Note that all functions have to be consistent with the chosen lookup table. A possible implementation to derive lookup tables and the required functions is the following. We iterate from the "right" side to the "left" side of the respective subsystem. If the local state at site $i$ is not maximal ($\neq |Q-1\rangle$) and the number of excitations $n_{\text{prev}}$ on previous sites is greater than one, we can increase the state at site $i$ and set the previous sites to the right of $i$ to its minimal state defined by $n_{\text{prev}} - 1$. This is obtained by setting the remaining excitations $n_{\text{prev}} - 1$ as much to the "right" as possible.

We further defined a "container class" that is in charge of the lookup table.

- `get_local_index`$(|\vec{\sigma}\rangle, k)$:
  Returns the subsystem index for subsystem $P_k$: $\text{index}_k(\vec{\sigma}^{(k)})$.

- `get_local_state`$(\text{index}, k)$:
  Returns the subsystem $|\vec{\sigma}^{(k)}\rangle$ on subsystem $P_k$ with $\text{index} = \text{index}_k(\vec{\sigma}^{(k)}) = $ `get_local_index`$(|\vec{\sigma}\rangle, k)$. This is the reverse function of the previous one.

---

**Algorithm 2:** Function (i): `get_index`

---

**Data:** $|\vec{\sigma}\rangle$
$\text{index} = 0$                                                   /* initializing variables */
$\lambda = 0$
**for** $0 \leq k < N$ **do**
    $n_k = $ `get_n`$(|\vec{\sigma}\rangle, k)$                           /* local particle number in $P_k$ */
    $i_k = $ `get_local_index`$(|\vec{\sigma}\rangle, k)$                /* index from the lookup table */
    $c_k = \text{offset}_k(n_k, \lambda) + i_k \cdot \text{stride}_k(n_k, \lambda)$      /* contribution of $P_k$ as defined in Eq. (13) */
    $\text{index} = \text{index} + c_k$
    $\lambda = \lambda + n_k$
**end**
**return** $\text{index}$;

---

---

**Algorithm 3:** Function (ii): `increment`

---

**Data:** $|\vec{\sigma}\rangle$
$\lambda = 0$
$\Gamma = 0$
**for** $1 \le j \le N$ **do**
    $k = N - j$                               `/* iterate backwards through all subsystems starting with the last */`
    $n_k = \texttt{get\_n}(|\vec{\sigma}\rangle, k)$
    **if not** `is_maximal`$(|\vec{\sigma}\rangle, k)$ **then**              `/* increase state while persevering the particle number `$n_k$` */`
        $|\vec{\gamma}^{(k)}\rangle = \texttt{increment\_local}\,(|\vec{\sigma}\rangle, k)$      `/* increase the state `$|\Psi\rangle$` locally on `$P_k$` within the sector `$n_k$` */`
        $|\vec{\gamma}^{(k+1,\dots,N-1)}\rangle = \texttt{get\_minimal\_state}\,(\Gamma, \lambda)$       `/* minimal state on `$P_{k+1}$` to `$P_{N-1}$` with length `$\Gamma$` and `$\lambda$`
        particles */`
        $|\vec{\gamma}\rangle = |\vec{\sigma}^{(0,\dots,k-1)}\rangle \otimes |\vec{\gamma}^{(k)}\rangle \otimes |\vec{\gamma}^{(k+1,\dots,N-1)}\rangle$
        **return** $|\vec{\gamma}\rangle$
    **else if** $\lambda > 0$ and $n_k < (Q-1)L_k$ **then**                     `/* increase state particle number in `$P_k$` */`
        $|\vec{\gamma}^{(k)}\rangle = \texttt{get\_minimal\_state}\,(L_k, n_k + 1)$      `/* get the minimal state on `$P_k$` with length `$L_k$` and `$n_k + 1$`
        particles */`
        $|\vec{\gamma}^{(k+1,\dots,N-1)}\rangle = \texttt{get\_minimal\_state}\,(\Gamma, \lambda - 1)$    `/* minimal state on `$P_{k+1}$` to `$P_{N-1}$` with length `$\Gamma$` and `$\lambda - 1$`
        particles */`
        $|\vec{\gamma}\rangle = |\vec{\sigma}^{(0,\dots,k-1)}\rangle \otimes |\vec{\gamma}^{(k)}\rangle \otimes |\vec{\gamma}^{(k+1,\dots,N-1)}\rangle$
        **return** $|\vec{\gamma}\rangle$
    **end**
    $\Gamma = \Gamma + L_k$
    $\lambda = \lambda + n_k$
**end**
**return** $\texttt{get\_minimal\_state}\,(L, n)$                     `/* the input state is maximal; return the minimal state */`

---

**Algorithm 4:** Function (iii): `get_state`

---

**Data:** `index`
$\lambda = 0$
$\Gamma = L$
**for** $0 \le k < N - 1$ **do**
    Determine $n_k$ s.t. $\text{offset}_k\,(n_k, \lambda) \le \texttt{index} < \text{offset}_k\,(n_k + 1, \lambda)$   `/* determine the correct particle number on `$P_k$` */`
    $\texttt{index} = \texttt{index} - \text{offset}_k\,(n_k, \lambda)$
    $i_k = \texttt{index}/\text{stride}_k\,(n_k, \lambda_k)$              `/* determine the local index in the particle number sector `$n_k$` */`
    $|\vec{\sigma}^{(k)}\rangle = \texttt{get\_local\_state}\,(i_k, k)$                      `/* reverse lookup table */`
    $\texttt{index} = \texttt{index} - i_k \cdot \text{stride}_k\,(n_k, \lambda_k)$
    $\lambda = \lambda + n_k$
**end**
$|\vec{\sigma}^{(N-1)}\rangle = \texttt{get\_local\_state}\,(\texttt{index}, k)$                      `/* last subsystem */`
$|\vec{\sigma}\rangle = \bigotimes_k |\vec{\sigma}^{(k)}\rangle$
**return** $|\vec{\sigma}\rangle$