# Peer review of "DanceQ: High-performance library for number conserving bases"

_SciPost Physics Codebases_

## Round 1 · Referee Report · Anonymous (Referee 1) · 2024-9-7

Strengths
2 - The solution proposed in the basis enumeration algorithm has immediate and wide application.
Weaknesses
2 - The sectioning is sometimes unclear.
Report
The majority of the paper, section III, is dedicated to outlining an algorithm for constructing lookup tables which is well-explained. I appreciate the connection to the most common sectioning of the system into two subsystems as a limit of the algorithm.
I question whether, assuming the paper's topic is the library, the submitted paper meets point 4 of the Scipost Physics Codebases acceptance criteria. A description of the "logic of its workings and its added value" is provided for the algorithm (in great detail in Section III and Appendix B), however, I had to spend several hours trying to understand the library itself from the otherwise rich and detailed documentation available on GitLab. For instance, even an example program for the code is not provided. Section III F exemplifies this problem. There, the authors briefly introduce the Operator class, notably in a tone disconnected from the rest of the paper. This section clearly concerns the library design and not the basis construction algorithm. Further, the State, Basis, Vector, or variants of SparseMatrix and ShellMatrix classes are never mentioned but appear important to the design of the library.
As a consequence of this tension, I see some minor internal inconsistencies. For example: at the beginning of section IV, the authors write that "for optimal performance, the required memory should not exceed [the L3 cache] limit" while in section IV-A they conclude that "the lookup and retrieval of the indices plays only a secondary role during the full matrix-vector multiplication." This observation to me somewhat invalidates the conclusions drawn from Fig. 4.
I ran some example programs provided in the repository and found that the installation process is generally straightforward however the library has important dependencies which need to be configured separately. The instructions are provided however CMake is not configured to attempt to find some libraries automatically. Some further minor observations are included in the following section.
Requested changes
1 - An explanation of the core design philosophy of the package and some example code should probably be added.
2 - I found the discussion about matrix-vector multiplication (MVM) very insightful albeit short, focusing on the memory requirements of the lookup tables. I would be interested in, e.g., more precise timing for a single MVM which the authors characterize as "only slightly dependent on $N$". The "several MPI threads" should also be changed to a number. It would also be interesting to know how MVM with fixed $L$ and $N$ scales with increasing the number of cores. An important example could be comparing the $L=46$ spin system for $N=2$ and $N=3$ regarding memory requirement and time consumption.
3 - In the discussion of Fig. 5, the authors note "a clear trend that larger systems and larger filling fractions require mode subsystems for ideal [lookup table] performance." Yet some outliers are noticed in particular for $Q>1$ for larger $L=32$ in disagreement with this general trend. Can this be explained somehow?
4- Sections III E and IV could be improved by clearer subsectioning. I.e. a single subsection does not make sense to me.
5 - The symbols used in equations should be checked, e.g., the pairs $q-Q$ or $c_0-c_{P_0}$ are sometimes used interchangeably. The caption of Fig. 6 contains several typos.
6 - The authors are aware that Krylow space algorithms are useful beyond (imaginary) time evolution and ground state searches even without storing additional wavefunctions (e.g., Phys. Rev. B 102, 054408), so I am surprised by this omission, e.g., when they write "a Krylov subspace which can used to perform time evolution or to compute ground states and excitations..., or other eigenvectors using spectral transformations."
Recommendation
Ask for minor revision
We sincerely thank Referee 1 for their thorough review, insightful comments, and positive feedback on our manuscript. We have carefully considered their suggestions and made substantial revisions to address the concerns raised.
We agree with the referee's main point that the initial version gave the impression that the manuscript primarily focused on the algorithm, leading to some structural ambiguities. Our goal was to highlight both the algorithmic advancements and the DanceQ library. To clarify this and improve the manuscript's structure, we have now dedicated a main section specifically to DanceQ. It discusses its core modules, usage, and performance. In particular, we agree with the referee that Section III F is misplaced and moved it to the appendix. We also added an example code in the appendix.
The referee further pointed out that the two performance sections about the algorithm itself and the matrix-free matrix-vector multiplication were unclear. In the revised version, we made this distinction more apparent. The algorithm's performance only considers the lookup, while the matrix-multiplication is a more applied performance test taking into account additional data coming from the wave function.
The referee further brought up that the library is dependent on utilizing its full potential. This is done by design as we do not aim to compete with advanced linear algebra packages, but we try to provide a link between these high-performance packages and a user-friendly interface. We hope that the Docker containers might simplify this process.
We are grateful for the referee’s constructive feedback, which has significantly strengthened our manuscript. All requested changes have been addressed in the revised version:
- We added a new section introducing the core modules of DanceQ.
- We have expanded the discussion on matrix multiplication by adding a new figure that illustrates the runtime as a function of the number of subsystems. The memory consumption for the lookup table in a large system with L=48 at half-filling can be calculated directly. For N=2, each MPI thread requires 128 MiB, whereas for N=3, it only requires 512 kiB. If memory allows for using N=2, we do not expect significant differences in runtime, as the overhead from large-scale MPI communication is the dominant factor. The primary benefit of using N=3 is the substantial reduction in memory usage. The referee also inquired about the number of MPI threads required to handle 46 spins. To compute the ground state energy of 46 spins in the zero-magnetization sector, approximately 256 nodes with 512 GiB of memory are necessary. We have added this detail to the manuscript.
- Unfortunately, we do not have an explanation for these outliers. We made this apparent in the revised text.
- We changed the sectioning accordingly.
- We have addressed the inconsistency.
- We have included the option of performing imaginary time evolution.

Author: Robin Schäfer on 2024-11-07 [id 4941]
(in reply to Report 4 on 2024-10-19)We sincerely thank the author for their detailed contributed report and valuable suggestions, particularly regarding the emphasis on spatial symmetries. We fully agree that incorporating additional spatial symmetries is essential for advancing numerical boundaries in cutting-edge diagonalization techniques. We plan to include this feature in the upcoming version of DanceQ, which we have mentioned now, together with SPINPACK and Xdiag in the conclusion. We also appreciate the recommendation to reference J. C. Bonner and M. E. Fisher in Proc. Phys. Soc. 80, 508 (1962), which we have now included. Additionally, we thank the referee for suggesting the SciPost style to expedite production upon acceptance.
The referee correctly noticed that our approach parallelizes over the target vector. For matrix-free matrix-vector multiplication, each thread processes its assigned portion of the input vector and sends the generated matrix elements to other threads. While our library supports OpenMP, its primary focus is MPI-based applications, where each thread is responsible for its vector segment; thus, we do not face any potential write conflicts. However, the referee is right that we must be cautious in the case of shared memory applications. In fact, by parallelizing over the target vector, we could potentially eliminate the need for the “atomic” construct currently used in the OpenMP implementation. We will consider this adjustment in future versions, though we do not anticipate significant runtime changes from this modification.

---

## Round 1 · Referee Report · Anonymous (Referee 2) · 2024-9-11

Report
I think the presented manuscript could be published in SciPost Physics Codebases after revision. Overall I think that the manuscript provides valuable insight for numerically oriented research groups dealing with spin or Hubbard systems. I have a few recommendations:
-
The authors use "particle number" in order to name the deviation from some basis state that could be termed vacuum. However, since they use spin models as examples most of the time I suggest to explain this a bit more carefully in the abstract and introduction.
-
In the introduction the authors mention state of the art numerical work in Refs. 30-32. I suggest to add some work of Kristel Michielsen who for a long time has been holding records in terms of number of spins for dynamical problems, that by the way can be treated in similar ways.
-
A proof of Eq. (7) was presented in [51], not only the result. I leave it to the authors whether they want to present their proof in Sec. A; maybe it is more insightful or more easily accessible.
-
Either above Algorithm 1 or in the discussion it should be mentioned that the scheme developed in [50] applies to mixed spin systems not just Q=2.
-
Discussion on page 14: The use of "half filling" for spin systems is again (see 1) unusual or misleading. At least, the authors should add that this is equivalent to M=0.
-
If the author want to refer to one of the most potent free programs in the field, they could refer to spinpack by Schulenburg that uses the method of Lin. It was e.g. used for finite temperature Lanczos calculations for a kagome system with N=42 spins addressing the full Hilbert space as well as for N up to 72 addressing few magnon spaces in order to describe magnon condensation related to flat bands. spinpack works in MPI/openMP hybrid mode.
Recommendation
Ask for minor revision
We thank Referee 2 for their careful reading and constructive suggestions. In addressing their requested changes, we hope that the revised manuscript will be approved for publication:
- We made the distinction between the Bosonic and spin languages clearer by an additional paragraph in the introduction.
- We are particularly thankful for your pointing us to Kristel Michielsen’s works, which we have included.
- We made it apparent that Ref. [51] proved the result.
- We have clarified that Ref. [50] applies the limit N=L in the case of Q\geq 2.
- We have changed the wording from half-filling to the zero-magnetization on page 14.
- We have added the SPINPACK software to our references.

---

## Round 1 · Referee Report · Anonymous (Referee 3) · 2024-9-12

Strengths
1- Efficient and elegant numerical library to generate and access qubit configurations with a fixed U(1) charge (e.g. magnetization for spin system) 2- Open source code 3- Very generic code able to tackle any local Hilbert space
Report
This numerical library is very generic and provides an implementation for both mappings between physical configurations and indices in the full list. Thus, it is a very useful tool for any ED-like code. As reminded in the paper, a solution was provided by Lin long ago by dividing the system in two parts. The main achievement here is to generalize this idea to an arbitrary decomposition as well as local Hilbert space $q$.
Basically, it is possible to divide the $N$ sites into various subsystems such that the lookup tables can fit in memory (obviously if one uses less subsystems, computations will be more efficient but this is more costly in memory).
I find the paper very well written and the results convincing.
For all these reasons, I recommend the publication in SciPost Physics Codebases.
Requested changes
1- In many applications, $q=2$ (qubit or spin 1/2) and it is possible to generate valid spin configurations using a Gray-like code, which should be mentioned. Moreover, since they are ordered by construction, finding the index of a given configuration can be done with a simple binomial search. Why is it written that it is "prohibitevely expensive" ? It seems to me that it could be done for large systems as well ?
2- Minor changes: there are missing capital letters in some names, e.g. "bose", "fermi", "hubbard" etc.
Recommendation
Publish (surpasses expectations and criteria for this Journal; among top 10%)
We sincerely thank Referee 3 for their thorough review and for their very positive feedback, including the recognition of our work as being within the top 10% of the journal.
- We thank the referee for pointing us toward Gray codes, which we have mentioned in the revised version, but we are currently not aware of how to generate number-conserving basis states using these codes. The referee is correct that it is possible to store all states and perform a binary search. In fact, storing the states (or their representatives) and performing a binary search becomes necessary when implementing generic spatial symmetries. This procedure is described in detail in our earlier publication: Phys. Rev. B 102, 054408. The advantage of not storing additional state information is twofold. First, it eliminates the need for a binary search, which scales with system size. Second, memory is usually the constraining factor and storing the states would increase the memory usage by 50% in a simple ground state search, where two wave functions are needed to be stored. After reconsidering this, we concluded that the phrase "prohibitively expensive" was overly strong and have revised it to simply "expensive."
- We have capitalized the words in the revised version.

---

## Round 1 · Referee Report · Anonymous (Referee 4) · 2024-10-19

Strengths
Weaknesses
Report
The main shortcoming that I see is the absence of spatial symmetries. For example, at the end of the discussion (i.e., on page 14) there is a comment that 120 TByte of main memory would be required for a spin-1/2 system with 46 sites. If this system has translational symmetry, it should be possible to reduce this memory requirement to below 3 TByte, i.e., a much more reasonable amount. Implementing spatial symmetries efficiently is in fact the bigger challenge than number conservation. ${\cal H}\Phi$ [41] may not use spatial symmetries either, but there are packages implementing these, such as SPINPACK, https://www-e.uni-magdeburg.de/jschulen/spin/ (already mentioned by Referee 2) and Xdiag by Alexander Wietek, https://github.com/awietek/xdiag. To be quite clear, I am not saying that inclusion of spatial symmetries is a condition for publication of the present manuscript and library, but I think that the authors should add more comments on this issue than the half sentence that they devote to it on the right column of page 1.
A more technical comment concerns the parallelization strategy outlined in sections II B and IV A. If I understood correctly, they propose to parallelize over target. I believe that parallelization over source would be the better strategy since a fixed target element avoids write conflicts. Given that the Hamiltonian is a Hermitean matrix, the two strategies are formally equivalent, but e.g. CPU cache performance can differ when looping over source or target. I encourage the authors to think about this issue, and at least reconsider their related comments.
One part that I really enjoyed reading is the historical Introduction. I nevertheless permit myself to propose complementing the well-known Ref. [24] by the older but lesser known publication by J. C. Bonner and M. E. Fisher in Proc. Phys. Soc. 80, 508 (1962).
A final detail: I believe that typesetting of the $\vert\vec{\sigma^{(0)}} \rangle$ in Eq. (11) should be corrected.
Requested changes
1- Expand comments on spatial symmetries. 2- Cite further exact diagonalization packages. 3- Revisit parallelization strategy. 4- Possibly add a citation of J. C. Bonner and M. E. Fisher, Proc. Phys. Soc. 80, 508 (1962). 5- Correct typesetting of $\vert \vec{\sigma^{(0)} }\rangle$ in Eq. (11). 6- I was wondering why the authors did not use the SciPost style file. This is not mandatory, but would help to appraise formatting and may speed up production.
Recommendation
Ask for minor revision

---

## Editorial Decision

unknown